human-computer interaction/computational biology

network science, smartphone, smartphone apps, human behaviour

**Author for correspondence:**
Liam D. Turner
e-mail: TurnerL9@cardiff.ac.uk

# Evidence to support common application switching behaviour on smartphones

Liam D. Turner[1,2], Roger M. Whitaker[1,2],
Stuart M. Allen[1], David E. J. Linden[3,4,5], Kun Tu[6],
Jian Li[6] and Don Towsley[6]

[1]School of Computer Science & Informatics, [2]Crime and Security Research Institute, [3]MRC Centre for Neuropsychiatric Genetics and Genomics, and [4]Neuroscience and Mental Health Research Institute, Cardiff University, Cardiff, UK
[5]School for Mental Health & Neuroscience, Faculty of Health, Medicine and Life Sciences, Maastricht University, Maastricht, the Netherlands
[6]College of Information and Computer Sciences, University of Massachusetts, Amherst, MA, USA

LDT, 0000-0003-4877-5289; RMW, 0000-0002-8473-1913;
SMA, 0000-0003-1776-7489

We find evidence to support common behaviour in smartphone usage based on analysis of application (app) switching. This is an overlooked aspect of smartphone usage that gives additional insight beyond screen time and the particular apps that are accessed. Using a dataset of usage behaviour from 53 participants over a six-week period, we find strong similarity in the structure of networks built from app switching, despite diversity in the apps used, and the volume of app switching. App switch networks exhibit small-world, broad-scale network features, with a rapid popularity decay, suggesting that preferential attachment may drive next-app decision-making.

## 1. Introduction

The smartphone has become a ubiquitous and disruptive device [1,2], with human engagement becoming prolific. Consequently, there has been increasing research interest in usage levels of such devices [3,4], the habits we exhibit in using them [4–7] and how we structure tasks [8,9]. A further aspect of usage that has received less attention relates to 'surfing' behaviour, namely where a user navigates between applications (apps) to consume content [6,10–12]. Each app provides particular affordances, with the user free to switch and interact between them, enabling the pursuit of cognitive stimulation [13] or other forms of fulfilment (e.g. internet addiction [14,15]). This motivates the research question of how application switching provides insight into a smartphone user's

latent behaviour, beyond existing approaches to characterize individual usage patterns. Examining this in the context of similarity and dissimilarity between individuals forms the focus of this paper. To the best of our knowledge, switching behaviour has not previously been represented as a network, with previous studies primarily focusing on frequency of app use over time [1,16–18] or short sequences of usage [6]. App switching networks are of potential interest from a number of perspectives, including human–computer interaction, cognitive psychology and network science.

Although smartphone users may engage different sets of apps and may also have individual differences in their usage patterns [1,8,11,18], over the longer term every user is constrained by time and the cognitive limitations of the human brain. Therefore, the abstracted structure of app switching could be universally governed, to some degree, by a combination of human cognitive functions, including memory and social cognition, alongside time. Over a sustained period, overall usage may reflect these constraints. We note that universal constraints and patterns have been discovered in other contexts that could also contribute to cognitive engagement with smartphone apps, such as attention and working memory [19], communication [20–23], maintaining relationships [24] and interests [25], web revisitation [26,27] and mobility patterns [28–30].

Therefore, we hypothesize that the structure of smartphone app switching behaviour exhibits invariant characteristics, despite differences in the specific apps that individuals engage with. In other words, the network of apps that users create through switching belong to a particular class of network, as seen in other aspects of human behaviour (e.g. [22,31]). However, assessing app switching behaviour is non-trivial as it requires detailed monitoring of an individual's smartphone usage over a prolonged period. We resolve this through a bespoke app to record smartphone interaction.

# 2. Methods

## 2.1. Dataset

The app switch networks in this paper are defined from data made available from the Tymer project [32,33]. The project developed an app to monitor Android smartphone interactions made by 76 participants over an eight-week period, as well as self-reports of mood and other lifestyle characteristics. Those participants with usage recorded for at least 75% of the period (six weeks) were selected for analysis ($n = 53$). We used the first 6 weeks of data for each participant, resulting in over 192 000 app switches. App switching behaviour for each participant involved taking the sequence of application window change events that took place while the screen was on, filtering out events where the user interacted with the system rather than specific apps, and when an event was followed by another within 0.5 s to mitigate the effects of accidental switching. From this, the resulting sequences were traversed to create the set of switches forming an *app switch network*.

## 2.2. Network construction

To formally specify an app switch network for a particular user, let $V$ be the set of all apps accessed by all participants over the 42-day period. The app switch network for participant $i$, is denoted $G_i = (V_i, E_i)$, where $V_i \subseteq V$ is the subset of apps (nodes) used by $i$, and a directed edge $(u, v) \in E_i$ denotes that at least one switch took place from app $u$ to $v$. The weight of an edge $w_{uv}$ denotes the number of switches from app $u$ to $v$ by participant $i$ during the observed period. We examine the hypothesis by considering: (i) the extent of network similarity, both with and without labelling of nodes by specific app (§2.3), and (ii) whether the app switch networks could be described as belonging to a particular class of network (§2.4).

## 2.3. Network similarity

### 2.3.1. Size, specific switches and structure.

We examine the similarity between the networks by comparing their size, the individual apps and switches they contain, and the connectivity structure independent of individual apps. Descriptions of the metrics used are defined in the appendix. Firstly, the number of nodes (apps), edges and switches allow differences in the volume of switching between individuals to be assessed. From this, the edit distance between the networks [34], defined as the proportion of node and edge changes needed for a given graph $G$ to become identical to another graph $H$, enables analysis of the changes needed for the

networks to contain the same apps and switches. Finally, the density and reciprocity allow the overall connectivity within the networks to be examined, with in+out degree, strength and centrality metrics providing further insight into whether this connectivity is balanced across the nodes. Network motifs enable further insight of the structure at a local level.

### 2.3.2. Network motifs.

Network motifs [35] characterize a network by considering the presence of induced subgraphs, relative to expectation from a null model. This gives a basis for network comparison. As the compared networks may be of different sizes, we use the subgraph ratio profile (SRP) as defined by Milo *et al.* [36] to represent the local triadic structure of each network as a vector. For each triad $i$, an SRP score ($SRP_i$) is calculated by firstly measuring the difference between the number of occurrences of $i$ observed in the network ($N_{ob_i}$) and the average number of occurrences in random networks produced by a null model ($\langle N_{rand_i} \rangle$), where

$$\Delta_i = \frac{N_{ob_i} - \langle N_{rand_i} \rangle}{N_{ob_i} + \langle N_{rand_i} \rangle + \epsilon}.$$  (2.1)

$\epsilon$ is an error term to ensure that $\Delta_i$ is not too large when $i$ rarely appears in either the assessed network or random networks of the null model. $\epsilon = 4$ is adopted [36]. From this, $\Delta_i$ is normalized against all triad $\Delta$ scores as $SRP_i$ [36], where

$$SRP_i = \frac{\Delta_i}{(\sum \Delta_i^2)^{1/2}}.$$  (2.2)

A large positive or negative value of $SRP_i$ indicates that a triad occurs much more or less frequently in a network than would be expected by random chance (i.e. a network motif or anti-motif [35,36]). To compare the similarity in the triadic structure between the app switch networks, we calculate the correlation coefficient [36] between each pair of networks. In the analysis, we use a null model that controls for random graph generation with the same bi-degree sequence in the results. However, similar results are obtained for alternative null models where the same number of nodes and edges are controlled for, and where the same number of mutual, asymmetric and null ties are controlled for.

## 2.4. Classification of network connectivity

To determine whether the structure of app switch networks are akin to a particular type of complex network, we perform goodness of fit assessment [37] on the in+out degree and node-strength distributions, setting $x_{min} = 1$. The distributions considered for fitness were power law, truncated power law and exponential, which, respectively, correspond to scale-free, broad-scale and single-scale classifications of small-world networks [38], where the distribution in graph connectivity follows a power law, truncated power law and exponential distribution, respectively. We determine significance of the fitness if a particular distribution is significantly better fitting than all others ($p < 0.05$, see §3).

# 3. Results

To investigate our hypothesis, we structure the analysis by firstly examining the extent of similarity between the app switch networks. From this, we explore whether this similarity can also attribute the networks to a particular class of network.

## 3.1. The extent of network similarity

Table 1 (top) shows considerable variation in the volume of app switching (node in+out degree, strength and total app switches) and the scope of switching between applications (number of nodes and edges). Figure 1 shows a moderate commonality in the presence of particular apps across the networks, however the edit-distance between the networks is high, indicating that the overall commonality of app placement and dominance is highly variable by individual user (figure 2*a*, mean (M) = 0.89, median (Mdn) = 0.86, standard deviation (s.d.) = 0.06, see §2.3), with similar values if direction or edge-weights are removed. Consistent with this, we find low commonality across each user's top five apps, defined by node strength (M = 32%, Mdn = 40%, s.d. = 19%), with similar

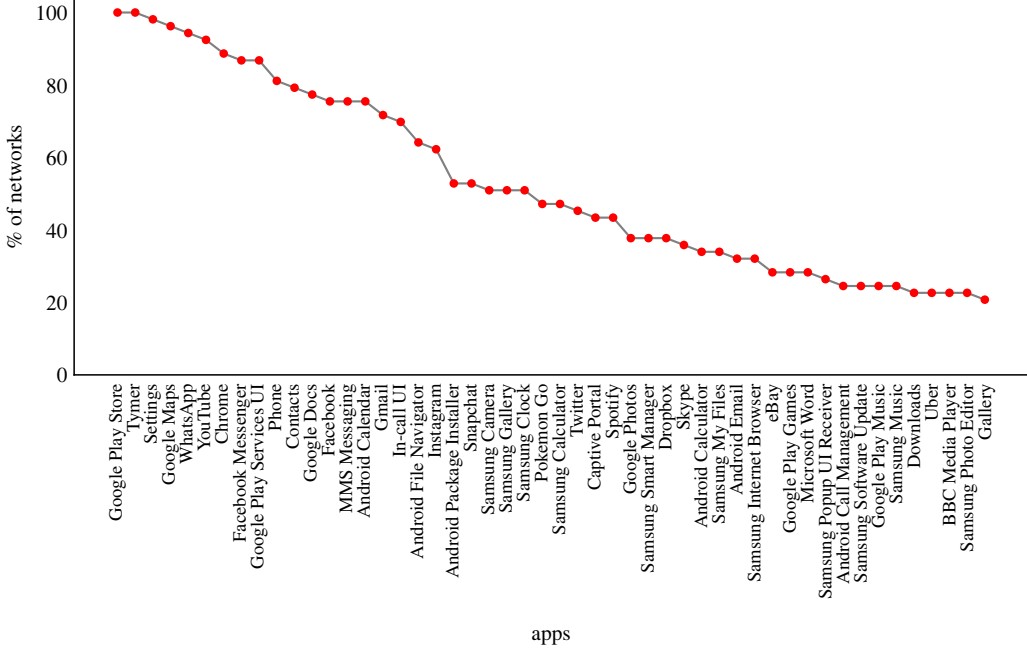

**Figure 1.** Frequency of the most popular apps across the 53 app switch networks.

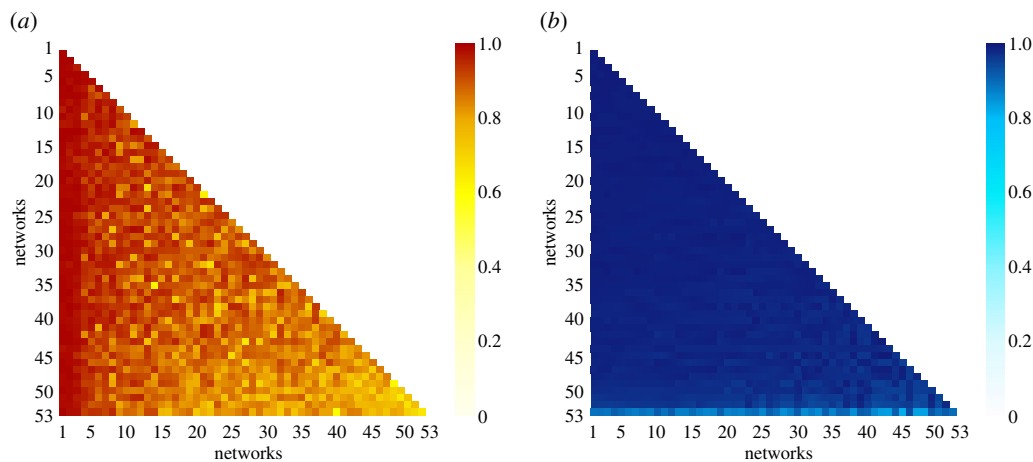

**Figure 2.** Pairwise comparisons of the app switch networks. (a) Similarity matrix based on edit distance [34] (0 = identical networks (white), 1 = completely dissimilar networks (red)). (b) Correlation coefficient matrix based on subgraph ratio profiles [36] (see §2.3) (0 = profile dissimilarity (white), 1 = strong profile similarity (blue)).

results if the set of all apps are considered. These findings are consistent with observations of individuality in terms of screen time and app usage from the literature (e.g. [1,4]), with some shared commonality in nodes likely due to the presence of popular applications (e.g. social media).

Despite this variability, the normalized statistics in table 1 (bottom) indicate possible structural similarity in app switching behaviour, consistent with the hypothesis. Note that these statistics are independent of network size. The underlying structural similarity extends to the local induced substructure of the app switch networks. Figure 2b shows that the pairwise correlation coefficient between the triadic SRP [36] of the networks is high, using a null-model of random graphs with the same bi-degree sequence ($M = 0.98$, $Mdn = 0.99$, s.d. $= 0.02$), with similar results for other null models (see §2.3).

In other words, similar relative frequencies of induced triadic subgraphs are seen, showing that the structural similarity observed at a global level (table 1) is not obfuscating diversity at a local level, adding further support to our hypothesis. Additionally, this indicates that the typical SRP of the networks could define app switch networks as a super-family [36] of local network structure. In particular, we note that

**Table 1.** Statistics for the 53 app switch networks $G_i$. Top: Statistics pertaining to network diversity. Bottom: Statistics pertaining to network similarity. See appendix for descriptions of statistics. M, mean; Mdn, median; s.d., standard deviation across the networks.

| statistic | M | Mdn | s.d. |
|---|---|---|---|
| number of apps ($|V_i|$) | 61.9 | 60.0 | 20.4 |
| number of edges ($|E_i|$) | 488.5 | 476.0 | 222.1 |
| mean degree | 15.3 | 14.6 | 3.7 |
| mean node strength | 116.3 | 101.9 | 66.9 |
| total app switches | 3636 | 3047 | 2355 |
| **normalized statistic** | **M** | **Mdn** | **s.d.** |
| density | 0.14 | 0.12 | 0.05 |
| reciprocity | 0.71 | 0.72 | 0.05 |
| weighted reciprocity | 0.77 | 0.76 | 0.09 |
| local reaching centrality | 0.92 | 0.94 | 0.06 |
| mean degree centrality | 0.27 | 0.24 | 0.09 |

**Table 2.** Clustering and path analysis suggesting small-world characteristics across the app switch networks. M, mean; Mdn, median; s.d., standard deviation across the networks.

| statistic | M | Mdn | s.d. |
|---|---|---|---|
| mean clustering coefficient | 0.59 | 0.6 | 0.08 |
| mean shortest path | 2.22 | 2.21 | 0.15 |
| mean betweenness centrality | 0.02 | 0.02 | 0.01 |

triads with at least one edge between all nodes, that also have one or more reciprocated edges, are typically over-represented in the network, with those uni-directional (e.g. feed-forward) or cyclic being commonly under-represented.

## 3.2. Small-world characteristics

The findings in table 1 and figures 1 and 2 show support for the hypothesis that app switching behaviour has invariant characteristics through the similarity seen in the global and local structure, despite notable differences in network size and the specific apps and switches they contain. This motivates further investigation of the typical structure of app switch networks, in order to compare the networks to common types of network structures. Smartphone users appear to be selective in the apps that they switch between (indicated by low density), but a path exists between most pairs (local reaching centrality). A user's switching is also highly reciprocal overall, although there are cases where this is imbalanced, indicative of particular app sequencing highlighted in previous studies [5,39], such as routines where a user typically uses one particular app before another. Additionally, there is minor disassortativity when examining whether apps in the networks with the same Google Play Store category are connected (M $= -0.03$, Mdn $= -0.03$, s.d. $= 0.03$), suggesting that navigation frequently crosses app categories.

The sparse, reciprocative, but connected structure, combined with imbalance in node degree and strength is suggestive of a small-world network structure. To investigate this, table 2 shows that the networks have a higher mean clustering coefficient (M $= 0.25$, Mdn $= 0.25$, s.d. $= 0.01$) and mean shortest path (M $= 2.14$, Mdn $= 2.14$, s.d. $= 0.04$) in comparison to 10 000 random Erdös–Rényi networks generated with the same number of nodes and density of edges, with similar mean betweenness centrality (M $= 0.02$, Mdn $= 0.02$, s.d. $= 0.01$). This suggests that the networks have small-world network characteristics. To classify this further, we apply goodness-of-fit analysis [37]

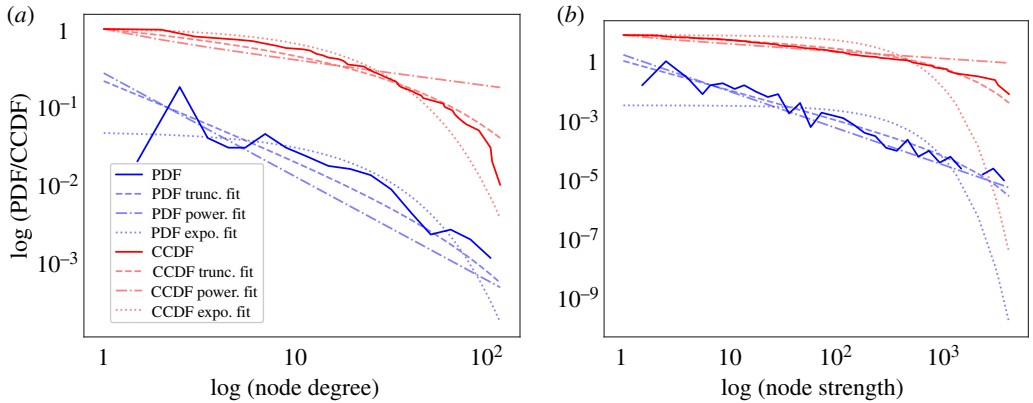

**Figure 3.** Distribution fitting [37] (see §2.4) for an example app switch network $G_i$ using a probability density function (PDF) and complementary cumulative distribution function (CCDF)—setting $x_{min} = 1$. (a) In+out degree sequence (power law $\alpha = 1.32$; truncated power law $\alpha = 1.0000001$, $\lambda = 0.01$; exponential $\lambda = 0.05$). (b) In+out node strength sequence (power law $\alpha = 1.25$; truncated power law $\alpha = 1.00001$, $\lambda = 0.001$; exponential $\lambda = 0.04$).

(see §2.4) on the connectivity distributions of the app switch networks to determine if they belong to a particular subclass of small-world network [38]; an example is shown in figure 3.

We find that the node strength [40,41] and in+out degree distributions of the app switch networks follow those of broad-scale and single-scale networks, respectively, suggesting that most user attention surrounds a small number of apps. For example, the second most used app in switches is used on average 73% as much as the most used app (Mdn = 0.77, s.d. = 0.19). The node strength distribution for 98% of networks ($n = 52$) best fits a truncated power law (broad-scale network) ($\alpha : M = 1.01$, Mdn = 1.0, s.d. = 0.03; $\lambda : M = 0.004$, Mdn = 0.002, s.d. = 0.02) which is significant ($p < 0.05$) [37] for 84.6% of networks ($n = 44$). In the remaining networks, an exponential distribution cannot be ruled out. For the in+out degree distribution, 90% of networks can be best described by an exponential distribution (single-scale network) ($n = 48$, $\lambda : M = 0.07$, Mdn = 0.07, s.d. = 0.03), with the remaining following a truncated power law as with node strength. However, the exponential fit is only significant for 39% of networks ($n = 19$), suggesting that while we have moderate support for an exponential distribution, the possibility of a truncated power law (or power law) cannot be ruled out. These results reaffirm inter-network similarity in support of the hypothesis, and provide a basis for comparison with other types of networks.

## 4. Discussion

The results establish that there is underlying similarity in the characteristics of app switch networks, despite substantial diversity in usage, such as the range of apps and volume of switching. This is evident in both the global and local structure (§3.1), as well as in the distribution of node connectivity (§3.2), where the node strength distribution exhibits strong fitting against a truncated power law, consistent with a broad-scale network. Arguably the node strength connectivity metric best captures app switching at the node level, as it combines the volume of switching an app is involved in.

These observations support the hypothesis that app switching is governed by a common characteristic. Critically, rather than maintaining a pool of similarly important apps through which switching occurs, it appears that humans exhibit a rapid drop off in switching activity to alternative apps. This rapid decay in popularity suggests that next-app decision making is driven by a preferential attachment mechanism. Preferential attachment is well known to support the formation of networks characterized by a power law distribution [42] and a truncated power law is evident in app switch networks. In other words, a highly popular app could gain additional switching from new apps, because they are already retained as popular, either consciously through memory or subconsciously through habit.

The results complement and extend existing knowledge concerning smartphone usage (e.g. [4–7]), which has focused on the differences between individual users' behaviour, as reflected by variations in our descriptive statistics of the networks. These remain important contributions, for example, in relation to a particular context or habit. However, by accumulating app surfing behaviours over a sustained period, we are aggregating and mediating factors such as context or timing, that may locally influence and skew individual usage. Potential individual differences (e.g. personality type)

across the user population clearly remain, and the results indicate that despite such individual differences, an underlying commonality in overall smartphone app navigation persists.

## 5. Conclusion

Smartphones have become an ubiquitous aspect of daily life, with previous studies showing that app usage is often diverse and individual (e.g. [1,5,43]). This paper has examined an additional aspect of smartphone usage behaviour that has received considerably less attention—how we switch between applications. We have introduced and analysed the app switch networks of 53 users over a 6-week period and find support for the hypothesis that independent from individual differences in the apps we use, and how frequently we use them [1,44], the structure of app switching has invariant characteristics between users.

From a network science perspective, the app switch networks show a small-world phenomena, having broad-scale characteristics. We note that this is consistent with a diverse range of human behaviour where network characteristics follow power law and truncated power laws, such as mobility [28,30,31], interests [25] and communication [22,45], as well as other application areas (e.g. [37,46,47]). We further hypothesize that our findings are the result of preferential attachment in decision-making. More broadly, the results give potential insights into the cognitive and temporal limitations in maintaining attention across smartphone apps. Albeit in a different context, we are aware that cognitive and temporal constraints have been established concerning attention for maintaining human social relationships (e.g. [24]).

The research has also highlighted that app switch networks may provide insights into smartphone users' latent behaviour, beyond existing approaches to characterize individual usage patterns. Because they effectively capture the structure of app popularity and present a map of common routes for app surfing, app switch networks could offer additional features for next-app prediction [16,17,39], for purposes such as displaying recommendations or pre-loading applications. Additionally, the presence of particular substructures, indicated through network motifs, may correlate to the psychological status of the user, noting that deviations in switching has been established as a useful proxy for mood [48].

Ethics. The anonymized dataset used in this work was originally produced by the Tymer project at Cardiff University. The Tymer project was approved by the ethics committee of the School of Psychology, Cardiff University (EC.16.04.12.4490) and was performed in accordance with the Declaration of Helsinki. All participants provided written, informed consent to participate and for their anonymized data to be made available.

Data accessibility. The data used for this study is available at: https://doi.org/10.5061/dryad.4v4bn15 [49].

Authors' contributions. L.D.T., R.M.W. and S.M.A. formulated the hypothesis for the study and the methodology. L.D.T. undertook network analysis of the data, supported by K.T., J.L. and D.T. who undertook the network motif analysis. L.D.T., R.M.W., S.M.A., D.E.J.L. and D.T. synthesized the results. All authors contributed to writing and reviewing the manuscript.

Competing interests. We have no competing interests.

Funding. This research was sponsored by the U.S. Army Research Laboratoryand the U.K. Ministry of Defenceunder Agreement Number W911NF-16-3-0001. The views and conclusions contained in this document are those of the authors and should not be interpreted as representing the official policies, either expressed or implied, of the U.S. Army Research Laboratory, the U.S. Government, the U.K. Ministry of Defence or the U.K. Government. The U.S. and U.K. Governments are authorized to reproduce and distribute reprints for Government purposes notwithstanding any copyright notation hereon.

Acknowledgements. We thank the Wellcome Trust for funding the Tymer project which produced the dataset used in this work, and the participants for undertaking that study.

## Appendix. Descriptions of network analysis metrics

In conducting this analysis for each network $G$, we use standardized metrics from network science literature to determine how similar or different the app switch networks are in terms of size, content and structure (e.g. connectivity, centrality and paths). In doing so, we use the following definitions:

*Mean degree:* the average number of edges (in+out) incident to a given node in $G$; *Mean node strength* [40,41]: the average sum of all edge weights (in+out) incident to a given node in $G$ (weighted in+out degree); *Total app switches:* the sum of all edge weights in $G$; *Density:* the proportion of edges in $G$ that exist, in comparison to a complete graph with the same number of nodes; *Reciprocity:* the proportion of dyads in $G$ with bi-directional edges; *Weighted reciprocity:* total reciprocated weight as defined by

Squartini *et al.* [40] in *G*; *Local reaching centrality:* the average proportion of other nodes to which a given node in *G* has a path (un-weighted); *Mean degree centrality:* the average of the proportion of other nodes a given node in *G* is connected to; *Mean clustering coefficient:* the average of the fraction of possible triangles through each node in *G* that exist; *Mean shortest path:* the average length of the shortest path between each pair of nodes in *G*, where a path exists; *Mean betweenness centrality*: the average normalized sum of the fraction of all-pair shortest paths that pass through each node in *G*; *Assortativity:* a Pearson correlation coefficient of whether nodes in *G* with the same attribute (in our case Google Play Store category if available) are connected.

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
