## [Reviewer comments · Royal Society Open Science]

Review History

RSOS-190018.R0 (Original submission)

Decision letter (RSOS-190018.R0)

18-Feb-2019

Dear Dr Turner:

It is a pleasure to accept your manuscript entitled "Evidence to support common application switching behaviour on smartphones" in its current form for publication in Royal Society Open Science.

Please note: before we can complete the processing of your paper for publication, we require the following:

1. An editable file version of your paper (Word, or Latex are ideal);
2. Individual files for your figures and tables;
3. A document including the captions for your figures and tables.

Reports © 2019 The Reviewers; Decision Letters © 2019 The Reviewers and Editors; Responses © 2019 The Reviewers, Editors and Authors. Published by the Royal Society under the terms of the Creative Commons Attribution License <http://creativecommons.org/licenses/by/4.0/>, which permits unrestricted use, provided the original author and source are credited

Please email these (as a zip folder is fine) to openscience@royalsociety.org as soon as possible -- as soon as we've these files, we can process the accepted manuscript, but we are unable to do so until you've sent them over.

on behalf of Dr Cecilia Mascolo (Associate Editor) and Professor Marta Kwiatkowska (Subject Editor).

Associate Editor comments to authors:

I have reviewed the topic of the paper and the reviewers' comments and subsequent replies. I think the paper is of interest to the open science audience and that the authors have made considerable efforts to answer the reviewers comments (which were mostly about positioning and motivation). I believe this should be published in open science.

Follow Royal Society Publishing on Twitter: [@RSocPublishing](https://twitter.com/RSocPublishing)
